# Design and Implementation of an Ontology for Measurement Terminology in Digital Calibration Certificates

**DOI:** 10.3390/s24123989

**Published:** 2024-06-19

**Authors:** Shuaizhe Wang, Mingxin Du, Zilong Liu, Yuqi Luo, Xingchuang Xiong

**Affiliations:** 1National Institute of Metrology, Beijing 100029, China; wangshuaizhe@nim.ac.cn (S.W.); liuzl@nim.ac.cn (Z.L.); 2Key Laboratory of Metrology Digitalization and Digital Metrology for State Market Refulation, Beijing 100029, China; 3The College of Information Engineering, China Jiliang University, Hangzhou 310018, China; mingxin@cjlu.edu.cn (M.D.); s21040803026@cjlu.edu.cn (Y.L.)

**Keywords:** digital calibration certificate, ontology, terminology for measurement, SI reference point, metrology digitalization

## Abstract

Digital Calibration Certificates (DCCs) are a key focus in metrology digitalization, necessitating that they satisfy the criteria for machine readability and understandability. Current DCCs are machine-readable, but they are still missing the essential semantic information required for machine understandability. This shortfall is particularly notable in the lack of a dedicated semantic ontology for measurement terminologies. This paper proposes a domain ontology for measurement terminologies named the OMT (Ontology for Measurement Terminology), using a foundation of metrological terms from standards like the International Vocabulary of Metrology (VIM), the Guide to the Expression of Uncertainty in Measurement (GUM), and JJF1001. It also incorporates insights from models such as the SI Reference Point, the Simple Knowledge Organization System (SKOS), and the DCC Schema. The methodology was guided by Stanford’s Seven-Step Method, ensuring a systematic development process tailored to the needs of metrological semantics. Through semantic expression capability verification and SPARQL query validations, the OMT has been confirmed to possess essential machine readability and understandability features. It has been successfully integrated into version 3.2.1 of DCCs across ten representative domains. This integration demonstrates an effective method for ensuring that DCCs are machine-readable and capable of interoperating within digital environments, thereby advancing the research in metrology digitization.

## 1. Introduction

With the ongoing digitalization across various industries progressing, fields such as Industry 4.0 [1,2] and smart manufacturing [3] are confronting increasingly complex challenges that demand enhanced technological support [4]. Metrology, critical to industrial operations, urgently requires a transformation to digital processes. In 2017, Germany’s Physikalisch-Technische Bundesanstalt (PTB) launched a strategic [5] initiative focusing on the digital transformation of metrological services, which included four key structured projects. By 2022, the International Committee for Weights and Measures (CIPM) identified metrology’s digital transformation as a crucial component of its 2030+ strategy [6]. Additionally, in November 2022, the 27th World Congress of Metrology agreed on a resolution concerning “Global Digital Transformation and the International System of Units” [7], marking a significant step towards standardizing metrological practices internationally.

Calibration is a crucial component of metrological activities, with its outcomes documented in calibration certificates. These are often available in paper format, or electronically as Word or PDF files [8]. Although these files are digitally stored, they consist of unstructured data and lack machine readability. As a result, other downstream systems can only reuse these data after they undergo manual processing or expert interpretation. Consequently, a key initiative in the digitalization of metrological services is the development of machine-readable Digital Calibration Certificates (DCCs) [9,10].

To facilitate the adoption of DCCs within metrology and industry, these certificates must be both machine-readable and understandable. Although current implementations using Extensible Markup Language (XML) satisfy the criteria for machine readability, they do not yet achieve machine understandability. Machine-understandable DCCs build upon machine readability, comply with the FAIR (Findable, Accessible, Interoperable, and Reusable) principles [11,12] which contribute to high quality of data management and stewardship, and support data exchange [13] across various systems while ensuring metrological traceability and reliability verification. Semantic representation of data and metadata is vital to make DCCs machine-understandable [14,15], bridging the gap between merely storing data and making them actionable and interpretable.

The semantic enhancement of DCCs depends on robust knowledge representation in the field of metrology. Ontologies, which formally represent concepts and relationships within specific domains [16], have proven to be a crucial tool for this purpose [13,17,18]. They employ a declarative approach to semantic modeling, effectively separating the semantic descriptions from application-specific logic. This enriches the data layer with meaningful semantic content and promotes greater interoperability [19] and reusability [20]. The W3C’s Semantic Web Best Practices and Development (SWBPD) Working Group has endorsed this methodology. Additionally, the CIPM has recognized the value of ontologies in supporting metrological concepts and the PTB also highlighted their advancements in DCC ontology development during the fourth DCC Conference [21], showcasing the significant role of ontologies in the digital transformation of metrology.

However, current ontologies in the metrology field focus on the semantic representation of quantities and units based on the International System of Units (SI) [22], such as the QUDT (Quantities, Units, Dimensions and Types) [23], QUDV (Quantities, Units, Dimensions, Values) [24], OM (Ontology of Units of Measure) [25], and the development of the vim ontology [26]. A few works concentrate on the development of ontologies for specific tasks [27,28,29]. This situation highlights a significant gap: the absence of a comprehensive, ontology-based vocabulary that spans the broader measurement field.

This paper proposes a fundamental ontology for the measurement domain named the OMT (Ontology for Measurement Terminology), a semantic model built on the Web Ontology Language (OWL) [30]. The OMT integrates concepts from metrology standards such as the International Vocabulary of Metrology (VIM) [31], the Guide to the Expression of Uncertainty in Measurement (GUM) [32], JJF 1001-2011 [33], and the SI Brochure [34], and draws from frameworks like the Simple Knowledge Organization System (SKOS) [35] and the SI Reference Point [36]. After semantic expression capability verification and SPARQL [37] query validations, the OMT v1.0 has been initially deployed in DCCs across ten metrological fields. The results show that the OMT comprehensively captures the calibration data and metadata within DCCs, infusing them with meaningful semantic content. This enhances the data’s interpretability and interoperability across machines and supports downstream systems at the service layer that rely on semantic data, proving the OMT’s effectiveness in facilitating a more connected and understandable digital metrology environment.

The structure of this article is as follows: The investigation of related work is detailed in Section 2. Section 3 introduces the methodology behind ontology engineering, detailing the comprehensive process involved in constructing an ontology. In Section 3.1, the ontology’s design is elaborated upon, including the definition of the conceptual domain and the principles governing ontology reuse. Section 3.2 discusses the specific implementation, covering the establishment of class inheritance structures, properties, and constraints. Section 4 focuses on validating the ontology, assessing it through its semantic expression capabilities and its machine readability. The testing of the ontology’s application on DCCs is outlined in Section 5, followed by a discussion of the test results in Section 6. Finally, we conclude in Section 7 with a summary of the findings and the overall contributions of the work.

## 2. Related Work

The digitization of metrology is advancing swiftly in areas such as scientific research, industrial applications, and public metrology services. A key focus within this digital transformation is DCCs. DCCs facilitate the automatic exchange and integration of data under unsupervised conditions, significantly enhancing the interoperability of instruments and measurement devices across different systems in the industrial sector. This capability for machine comprehension and interpretation is central to Semantic Web technologies, typically implemented via the Resource Description Framework (RDF) [38] and the OWL. The RDF, fundamental to the Semantic Web, addresses the limitations of XML’s tree structure by using SPO (Subject, Predicate, Object) triples to represent real-world concepts within the digital realm, thus forging semantic links between humans and computers. The OWL, with its higher level of abstraction, uses an object-oriented approach to establish hierarchical relationships between entities and supports logical reasoning based on first-order logic. Thus, in metrology, ontology is crucial for enabling interoperable metrological data exchange between humans and machines, and among various systems.

Numerous studies have established ontologies for quantities and units within metrology, with key contributions highlighted in Table 1. The QUDT framework, developed for the NASA Exploration Initiatives Ontology Models (NExIOM) project, aims at standardizing measurement units, quantities, dimensions, and data types, providing a comprehensive understanding of basic metrological concepts. QUDV constructs unit and quantity systems for system modeling, encompassing definitions of quantities, units, dimensions, and values. The OM framework outlines classes, instances, and properties related to quantities, units, and dimensions, suitable for diverse applications. While the QUDT, QUDV, and OM extensively address the concepts of quantities and units, their specific application focus results in a lack of adherence to broader metrology standards and norms. The vim ontology draws on various metrological standards to define quantities and units and further details concepts such as quantity values and scales, facilitating conversions between international and non-international units. However, it does not encompass fundamental metrological concepts like measurement results, measurement errors, and measurement uncertainties. The SI Reference Point, issued by the International Bureau of Weights and Measures (BIPM), represents the authoritative digital reference for the SI, enforcing metrological standards to a high degree but still omitting key general measurement concepts.

However, the DCC Metadata Schema encompasses more than just the concepts of quantities and units; the articulation of measurement results is equally crucial. DCCs’ measurement results encapsulate several fundamental metrological concepts such as measurement results, measurement uncertainties, and measurement conditions. Consequently, there is a need for an ontology specifically tailored for DCCs to enhance their interoperability. During the fourth DCC Conference, the PTB demonstrated their developed DCC ontology [21]. Unlike traditional domain ontology construction methods, this ontology was derived by mapping XML fields. This map-based ontology, which functions as either an application or task-oriented ontology, does not define concepts universally in line with metrological standards and norms. As a result, it might lack suitability for interactions with other systems.

Overall, existing metrology ontologies cannot cover concepts beyond quantities and units in the measurement field, such as QUDT, QUDV, OM, and vim ontology, which include basic concepts of quantities and units. The SI Reference Point provides authoritative references and services of SI, but it also lacks certain core concepts, such as measurement results, measurement uncertainty, the probability distribution of standard uncertainty, and the correlation coefficient of input quantities, etc. The findings highlight a significant gap in metrology: the absence of an ontology that is both machine-readable and interpretable, centered around fundamental measurement terminology, which is essential for supporting the interoperability of DCCs. The OMT proposed in this study is designed to encompass the fundamental concepts in the field of measurement, thereby addressing this deficiency. This allows for the generation of DCCs that are machine-readable and understandable.

## 3. Methodology for Ontology Development

Ontologies are classified according to their scope and domain specificity into categories such as top-level, domain, task, and application ontologies. The OMT is categorized as a domain ontology. Its construction employs Stanford’s Seven-Step Method [39], and the process is detailed in Figure 1. To guarantee the authority and universality of metrology terminology, the process begins by manually extracting basic concepts, such as essential parts of the measurement process, measurement results, measurement errors, and measurement uncertainties, from metrological standards and guidelines. Following the completion of knowledge extraction, the domains of these concepts are defined, adhering to the foundational principles of classifying classes and properties. Some concepts represent physical entities, like measuring instruments, while others describe relationships between entities, such as the characteristics attributed to these instruments. The next step involves delineating the hierarchy of classes and properties, considering both theoretical and practical applications. Once the classes and properties are established, the ontology is further refined with conceptual constraints to improve its machine understandability and interpretability.

Like industrial software development, ontology construction is a dynamic, iterative process driven by feedback. Therefore, validating the ontology is crucial to gather insights for enhancements. After it successfully passes two stages of validation, the initial version of the OMT can be realized. This dual validation process ensures that the ontology meets both technical and user-centric criteria, making it robust and effective for its intended applications.

### 3.1. Contextual Design of Ontology for Measurement

The contextual design of the OMT is illustrated in Figure 2. Considering the critical importance of quantities and units in metrology, along with foundational ontology-building principles, it incorporates the SI Reference Point, which serves as the authoritative digital reference for SI. The SI Reference Point provides Permanent Digital Identifiers (PIDs) for named SI units, prefixes, and defined constants, along with tools for parsing compound units. Additionally, to meet data and metadata expression standards, it leverages the SKOS and DCMI Metadata Terms, facilitating integration with other compatible metadata vocabularies in the context. As a conceptual model for metrology, the OMT extensively draws on the terminology, concepts, and relationships covered in the VIM, GUM, and JJF 1001. In the context of class inheritance hierarchies, the official documents of the VIM, GUM, JJF 1001, and SI Brochure provide essential guidance. These documents play a crucial role in ensuring that the ontology is applicable across diverse business systems by offering standardized definitions and frameworks that align with global metrological practices. This standardization facilitates interoperability and consistent understanding across different implementations.

### 3.2. Implementation

The foundational elements are classes, data properties, and object properties in the development of ontology. The design of classes involves establishing clear hierarchical and inheritance relationships between concepts, which is essential for articulating the structure of these concepts. Properties should adhere to metadata and information standard specifications to fully convey the concepts’ semantic information. This structured approach ensures that the ontology can communicate the intended meanings and relationships inherent to the domain it represents.

#### 3.2.1. The Inheritance Hierarchy of Classes

For the OMT, the hierarchical structure of classes is detailed in Table 2. This table highlights the parent classes along with their respective subclasses, illustrating the foundational concepts at the core of the ontology. The division of class hierarchies is based on two core principles: composition and inheritance. In composition, for example, measurement conditions such as Temperature and Humidity each form a subclass under the broader category of Operating Condition. In the principle of inheritance, specific types of uncertainty, like Expanded Uncertainty, are considered subclasses of Measurement Uncertainty. The representation of inheritance relationships in the ontology utilizes RDF SPO triples. For instance, the relationship where Expanded Uncertainty is a subclass of Measurement Uncertainty is articulated using RDF syntax. This can be expressed in the following RDF SPO triple format:omt:ExpandedUncertainty rdf:subClassOf omt:MeasurementUncertainty

#### 3.2.2. Definition of Properties

Properties within an ontology are categorized into data properties, object properties, and annotation properties. The type of a property is determined by its value range: as shown in Table 3, data properties link an instance to specific numeric, literal values or are defined by built-in ontology types like xsd:string. Object properties, on the other hand, are detailed in Table 4, which outlines the relationships between instances, helping to organize concepts into coherent groups that reflect the interactions and structures involved in a measurement process.

#### 3.2.3. Definition of Constraints

Establishing constraints on classes through relational definitions enables the construction of a model that illustrates the interconnections between core concepts. This method involves defining specific relationships that restrict and define how classes interact within the ontology, effectively mapping out the structural dynamics of the model. In the OMT, the essential relationships around *Measurement Result* are illustrated in Figure 3. In this figure, instances defined by the OMT are denoted by purple rectangles. Quantities are indicated in light blue, while orange rectangles are used to signify quantity values. Literal values are marked in green. Prefixes, in this context, are shown in pink, and units are represented using yellow. These results are extensively documented with metadata that cover aspects like the object that was measured, the uncertainty and its components, the measurement model, and the measuring instruments. To optimize the ontology, the values and cardinalities of the concepts and properties have been rigorously defined and constrained. This adjustment has significantly improved the model’s machine readability and its capacity for semantic interoperability. The primary concepts and their specific constraints are detailed in Table 5.

Measurement results are pivotal components of a calibration report, typically comprising the measured quantity value and its associated measurement uncertainty. The format for expressing quantity value, as dictated by metrological standards, involves three distinct elements:Value.Prefix.Unit.

Moreover, measurement uncertainty itself represents a quantity value but encompasses additional data, including uncertainty components or coverage factor. It is crucial to identify the sources of these components, whether they stem from the measuring instrument or the measurement model. The measurement results should also clearly delineate the quantity being measured, the utilized measurement model, and the instruments.

The chosen structure is designed with flexibility and scalability, allowing for continuous optimization of content and structure based on feedback. The design of an ontology, grounded in the principles of Object-Oriented Programming (OOP), facilitates the convenient definition, addition, removal, and modification of concepts. For instance, if upon implementation of the ontology it is discovered that only the concept of standard combined uncertainty is present, with expanded uncertainty being absent, it is possible to enrich the ontology with this missing concept in a straightforward and adaptable manner:omt:ExpandedUncertainty rdf:subClassOf omt:MeasurementUncertainty

This structured representation employs RDF SPO triples to articulate relationships between these concepts, creating a graph that enables logical reasoning and data exchange on a knowledge graph (KG) configured following this schema. This allows machines to interpret and manipulate data effectively within this structured framework.

## 4. Validation

To assess the semantic interoperability and machine readability of the OMT, evaluations will be conducted from two aspects: semantic expression capability and SPARQL query performance. Semantic expression capability assesses whether the ontology can accurately and formally represent the fundamental concepts of the metrology field in a universally applicable manner while SPARQL queries are utilized to verify the ontology’s effectiveness at the machine-readable level, ensuring that it can be efficiently accessed and understood by computational systems.

### 4.1. Semantic Expression Capability Verification

#### 4.1.1. Model for the Measurement Process of the Volume of a Given Cylinder

Instance validation is an effective approach for verifying the semantic expression capabilities of an ontology, where expressing a given instance can validate the ontology’s efficiency and applicability. As shown in Figure 4, the OMT has established a generally applicable model for the measurement process of the volume of a given cylinder. In this model, rectangles symbolize specific instances within the ontology’s hierarchical structure; ellipses denote the Quantity Value which includes Value, Prefix, and Unit; and diamonds indicate Quantities and definitions being anchored to the reused SI Reference Point. The primary components of the measurement process include Measured Object, Measurand, Measuring Instruments, Measurement Results, and Measurement Uncertainty, where measurement results consist of the Measured Quantity Value and Measurement Uncertainty, and the Uncertainty Components are linked to their sources through object properties.

In this case, there are two forms of expressing measurement results. Both forms attribute the Measured Quantity Values to the same instance. However, their approaches to detailing uncertainty differ; one uses combined standard uncertainty, while the other employs expanded uncertainty. Expanded Uncertainty is derived from the Combined Standard Uncertainty coupled with a coverage factor. This linkage allows for the measurement results expressed in extended uncertainty to be traced back to the three individual components of the combined standard uncertainty, thereby maintaining continuity in understanding the source of the uncertainty.

In a measurement process, the roles of the measuring instruments, measurement methods, procedures, and models are crucial. For instance, in this scenario, the micrometer is characterized by three key properties: resolution, indication error, and instrumental uncertainty. These attributes are vital parts of the metrological traceability chain, ensuring the reliability of measurement results. Additionally, measurement methods and procedures enhance the metadata associated with a measurement process, adhering to the FAIR principles of data management. The measurement model used identifies two input quantities, height and diameter, and one output quantity: volume. It also provides the calculation formula for volume, enabling machine readability and the translation of these data into machine language, facilitating automated processing and analysis.

#### 4.1.2. Expression of Probability Distribution

When assessing uncertainty components using Type B evaluation, it is necessary to estimate their probability distribution or suggest distribution assumptions to obtain the standard uncertainty. Suppose the estimated value of the measured variable X is x. If x is influenced by multiple independent factors of a similar magnitude, it would be reasonable to assume a normal distribution. However, if based on relevant data, the probability of *x* falling within the interval $\left(x-a,x+a\right)$ is 1, and the likelihood of it appearing anywhere within this interval is equal, then x follows a rectangular distribution. It is crucial for a proper Type B evaluation of standard uncertainty to explicitly state the probability distribution of the components.

In Figure 4, the uncertainty component $u_{3}$ of the combined standard uncertainty $u_{c}$ arises from the micrometer’s indication error. If the estimated value x falls within the interval $\left(x-0.01,x+0.01\right)$ with a probability of 1, and the likelihood of it appearing anywhere within this interval is equal, then *x* follows a rectangular distribution, and its standard uncertainty is:$$u_{x}=\frac{a}{\sqrt{3}}$$

The information above is represented using the OMT as depicted in Figure 5. The Type B uncertainty component $u_{3}$ stems from the indication error and follows a rectangular distribution. The distribution interval is determined by the upper and lower limits, thereby ensuring that the standard uncertainty can be traced back to the probability distribution information of the associated components.

#### 4.1.3. Consideration of Correlation Coefficient

It is noteworthy that the previous example presumes zero correlation between each input. However, when the correlation between inputs or uncertainty components cannot be ignored, it is necessary to determine the correlation coefficients between each component before combining the uncertainties.

For instance, consider a linear correlation relationship. The correlation strength between two measurements is indicated by the correlation coefficient. Depending on the specific circumstances, different methods can be used to determine this correlation coefficient. If the correlation coefficient is calculated according to its definition, then:$$\rho =\frac{K_{\xi \eta }}{\sigma_{\xi }\sigma_{\eta }}$$

In the formula, $K_{\xi \eta }$ signifies the covariance between the inputs, while $\sigma_{\xi }$ and $\sigma_{\eta }$ represent the standard deviations of $x_{\xi }$ and $x_{\eta }$. As depicted in Figure 6, the correlation coefficient’s information can be represented using the OMT. Assume two inputs, $x_{\zeta }$ and $x_{\eta }$, with their corresponding standard deviations being $\sigma_{\xi }$ and $\sigma_{\eta }$, and their correlation coefficient being ρ. In the OMT, the inputs are instances, and the standard deviation and covariance values are expressed using the literal value. In this figure, the purple rectangle signifies an instance in the OMT, the green rectangle indicates the literal value, and the instances are interconnected through relationships. This figure demonstrates the fundamental model for expressing the correlation coefficient in the OMT.

When the correlation coefficient is not obtained through calculation, the correlation coefficient of two input quantities can also be directly specified, expressed in the format of RDF’s SPO triples:$$\mathrm{omt}:\mathrm{Input}\_1\;\mathrm{omt}:\mathrm{hasCorrelation}\;\rho_{12}$$
$$\mathrm{omt}:\mathrm{Input}\_2\;\mathrm{omt}:\mathrm{hasCorrelation}\;\rho_{12}$$

This process underscores the OMT’s ability to systematically manage measurement data and its proficiency in clearly articulating the relationships and hierarchies among measurement elements, showcasing its capacity to formally express universally relevant measurement concepts.

### 4.2. SPARQL Query Validation

The SPARQL query language operates on the principle of subgraph matching, positioning it as a foundational technology of the Semantic Web. Ontologies are constructed with the RDF, which uses an abstract model of SPO triplets to represent the real world accurately. By employing SPARQL queries to validate ontologies, it is possible to ensure that ontologies not only adhere to the expected standards of data structure but also maintain consistency across different information systems.

The scenario depicted in Figure 4 led to the generation of an OWL file, analyzed using the SPARQL plugin within Protégé 5.5.0 [40], following the code outlined in Appendix A, Algorithm A1. This approach was applied to trace the sources of combined standard uncertainty in measurement results, specifically from the uncertainty components related to the measured quantity. The findings, detailed in Table 6, confirm that the instance modeled in the OMT is machine-readable and adheres to the principles of metrological traceability. This ensures that the origins of uncertainty components can be accurately identified from the measurement data. Details of the tables, the SPARQL queries used, and the results have been published at https://github.com/QilinCoding/OMT (accessed on 17 May 2024), more details can be found in Appendix B.

## 5. Application Testing

### 5.1. Motivation

For automated data exchange between humans and machines, or solely between machines, without any supervision, each part of the measurement process must be expressed in an unambiguously and universally understandable way. This ensures that all relevant metrological concepts can be communicated at the semantic level. DCCs enable such semantic exchanges across different measuring devices, playing a pivotal role in advancing sectors like Industry 4.0, future factories, and smart manufacturing. By employing the OMT to annotate DCCs, we can enhance their semantic expression, making these certificates not only readable but fully comprehensible and interoperable across various technological platforms.

### 5.2. Application Workflow

In response to the need for platform independence across various measurement devices, the ontology management framework utilizes Owlready2, a Python-based tool known for its lightweight quality and superior portability compared to the Java-based Jena framework. As illustrated in Figure 7, the OMT processes a DCC file, extracting and then outputting relevant semantic information. In this model, the relationships and constraints defined within the OMT guide the precise designation of entities for the data encapsulated in XML fields. This approach ensures that each piece of data is linked to clear semantic information, enhancing the overall interpretability of the data. By explicitly mapping XML fields to specific entities based on the OMT framework, the model creates a structured semantic layer that aids in data processing and integration. To preserve the DCC’s original format, this semantic information is encoded into a Base64 string and embedded into specific fields of the DCC XML file. This method ensures the XML file encapsulates both the original DCC content and the enriched measurement concepts from the OMT, maintaining compliance with DCC Metadata Schema. Consequently, such XML files can seamlessly serve as inputs for additional service-layer tasks, enhancing data integration and usability across systems.

### 5.3. Coverage Testing

Coverage testing is crucial due to the diverse ways in which different domains describe measurement results, which affect the measurement concepts present in DCCs from those domains. After producing semantically enhanced DCCs as outlined in Figure 7, concept coverage testing was conducted. This testing is vital for analyzing the scope of concepts addressed by DCCs in various fields and for identifying areas where the ontology may need refinement. For this test, measurement results from DCCs in ten representative metrology domains were selected, and both the total number of concepts present and the number of concepts that the ontology could cover were counted. The coverage rate calculated from these figures serves as an indicator of the ontology’s effectiveness in encompassing the diverse measurement concepts across domains.

## 6. Results and Discussion

The results of coverage testing, as detailed in Table 7, show that despite the considerable differences in the concepts and terminology across various domains, the OMT successfully covers most of the essential concepts. This comprehensive coverage confirms the OMT’s effectiveness in providing robust model support for the semantic enhancement of DCCs. This capability ensures that DCCs are interoperable across different systems in practical applications.

Compared with the QUDT and OM, which can only include information related to quantity and units, the OMT can include all the concepts. For example, for a real quantity in a DCC:
 <si:real>
        <si:label>angular frequency</si:label> 
        <si:value>39.58</si:value>
        <si:unit>\second\tothe{-1}</si:unit>
        <si:expandedUnc>
            <si:uncertainty>3.14</si:uncertainty>
            <si:coverageFactor>2</si:coverageFactor>
            <si:coverageProbability>0.95</si:coverageProbability>
            <si:distribution>normal</si:distribution>
        </si:expandedUnc>
 </si:real>

The aforementioned information encompasses value, unit, and expanded uncertainty. Furthermore, expanded uncertainty involves concepts like coverage factor, coverage probability, and probability distribution. Ontologies like the QUDT and OM can denote the unit $s^{-1}$. However, the OMT goes beyond expressing just the unit; it can also convey information about expanded uncertainty, coverage factor, coverage probability, and probability distribution.

Upon detailed comparison, it has been noted that the OMT does not cover certain concepts primarily related to administrative data. This oversight highlights specific areas where the OMT could be improved. By leveraging feedback from practical application cases, we can identify ways to further enhance this method. To enhance the ontology’s comprehensiveness, integrating additional top-level and domain-specific ontologies such as MetaData4Ing, which specialize in the management of data concepts, could effectively fill these gaps. Additionally, it might be beneficial to consider incorporating the PIDs provided by the SI Reference Point within DCCs, and integrating it into the D-SI metadata model to address existing deficiencies.

Future work will concentrate on improving the usability and practicality of the OMT. Key efforts will include developing user-friendly interfaces, such as graphical user interfaces and publicly accessible SPARQL query endpoints, which will enhance the ease of use of the model. To bolster its practicality while maintaining interoperability and reusability, there will be a focus on creating universal APIs for systems and programs and offering deployment hosting services.

## 7. Conclusions

Developing an ontology for metrological terminology is a vital strategy for facilitating the digital transformation of metrology. Current ontologies primarily address quantities and units but fall short of providing a structured representation of universally applicable measurement terms. This study proposes the OMT, a semantic model designed to meet the evolving semantic demands of metrology’s digital transformation. Specifically, the OMT enhances DCCs by improving their machine understandability and enabling semantic interoperability.

The development of the ontology primarily adheres to Stanford’s Seven Steps, a methodology tailored for building domain ontologies. This process starts by extracting relevant knowledge from metrological standards, including the VIM, GUM, JJF1001, and SI Brochure. Based on this information, a conceptual domain is defined, organizing knowledge into a structured format. Subsequent steps involve sorting these concepts, initially without hierarchical order, into defined classes and properties, and applying necessary constraints. This structured approach culminates in a semantic model that articulately represents the fundamental concepts of the measurement field, enhancing clarity and interoperability within related applications.

We conducted a thorough assessment of the OMT to verify its capabilities in semantic expression and SPARQL querying. This evaluation aimed to confirm that the OMT accurately represents the measurement process formally and is effectively readable and interpretable by machines. The validation process included applying the OMT to version 3.2.1 of a DCC. Additionally, we tested the coverage of concepts within DCCs across various metrology domains, ensuring that the ontology aligns well with practical needs and standards in the field. This step was critical in demonstrating the OMT’s applicability and effectiveness in real-world scenarios.

Validation and experimental results confirm that the OMT effectively formalizes metrological concepts, offering robust support for enhancing the semantic representation of DCCs. The OMT aligns with established metrological and information standards, fulfilling the essential needs for machine understandability and semantic interoperability in DCCs. This capability facilitates the precise and unambiguous exchange of data and metadata across the domain, ensuring that information is consistently interpreted and applied. This enables the ontology to interpret the measurement data in the DCCs without ambiguity, and flawlessly convey metadata concerning measurement, conditions, and data quality. For instance, in a typical Industry 4.0 environment, a DCC annotated by the OMT can be automatically read, analyzed, interpreted, and processed by machines across the entire workflow chain. This reduces labor costs, enhances the operational efficiency of equipment, and consequently, yields economic benefits for automated factories.

Furthermore, this development aids in building a worldwide consensus on metrological concepts, providing robust, intuitive, and practical interface support for service-layer applications. Such advancements are essential for advancing the digital transformation of metrology.

## Figures and Tables

**Figure 1 sensors-24-03989-f001:**
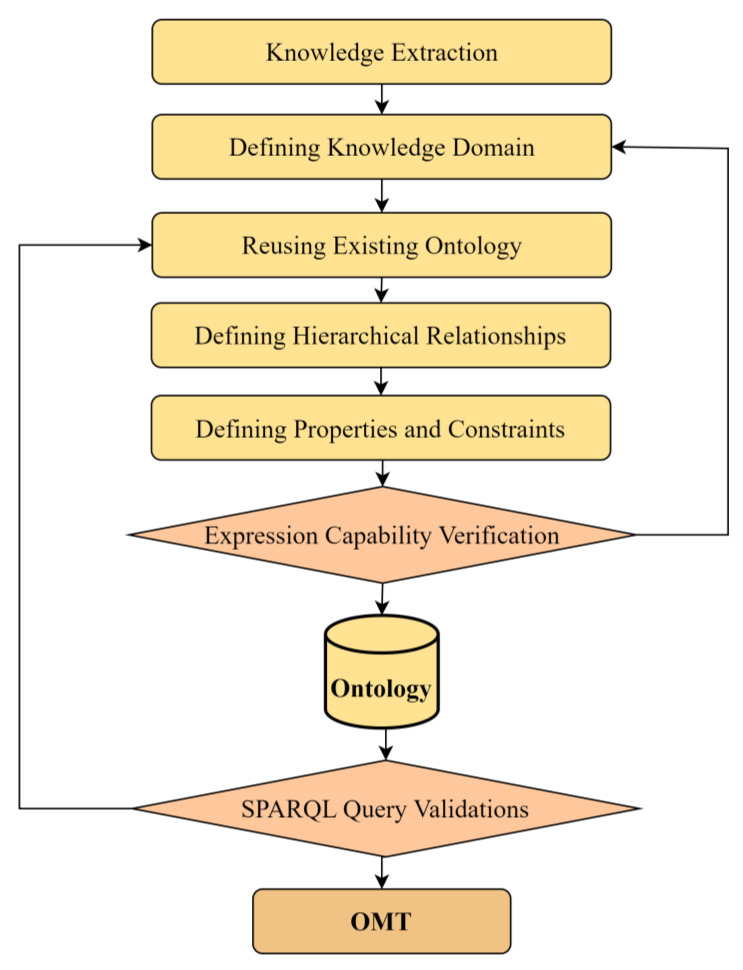
Construction workflow of Ontology for Measurement Terminology (OMT).

**Figure 2 sensors-24-03989-f002:**
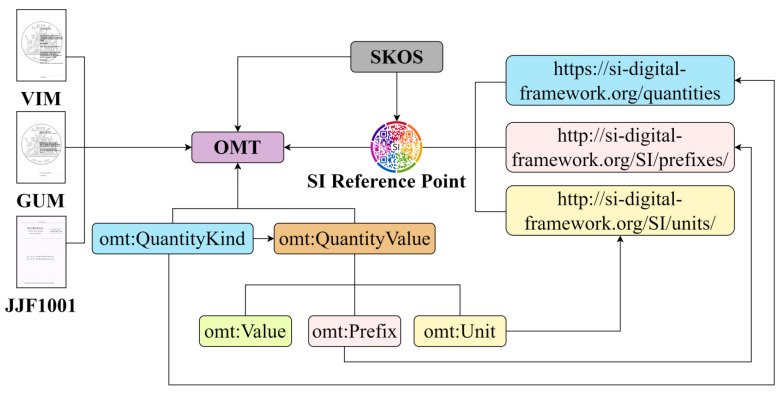
Contextual relationships of OMT(The JJF1001 represents the General Terms in Metrology and Their Definitions of China).

**Figure 3 sensors-24-03989-f003:**
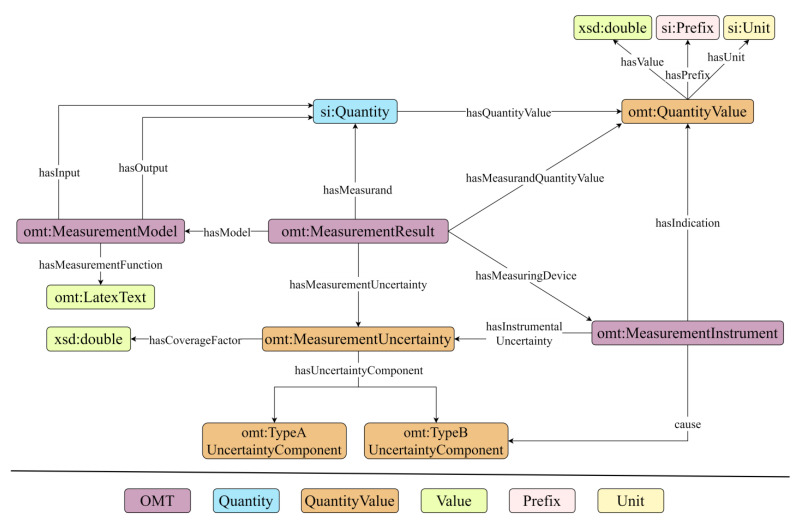
Relationships of core concepts.

**Figure 4 sensors-24-03989-f004:**
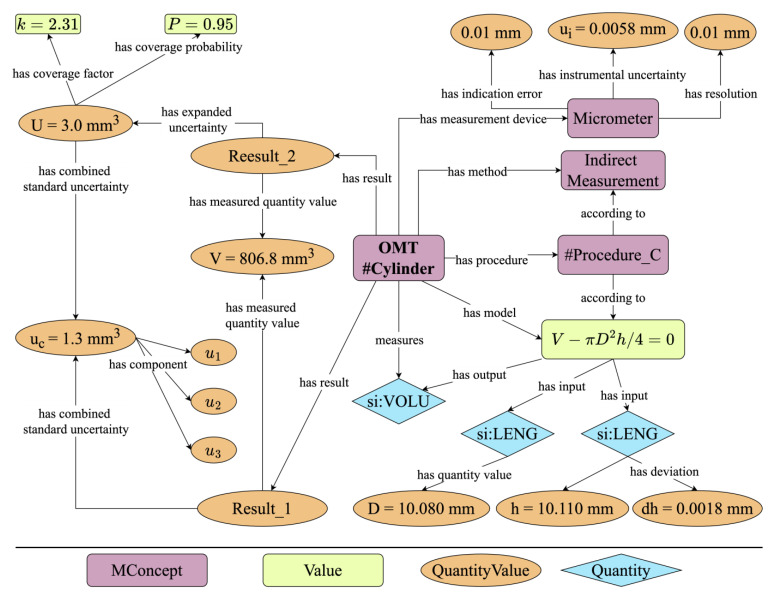
Measurement process of the volume of a given cylinder.

**Figure 5 sensors-24-03989-f005:**
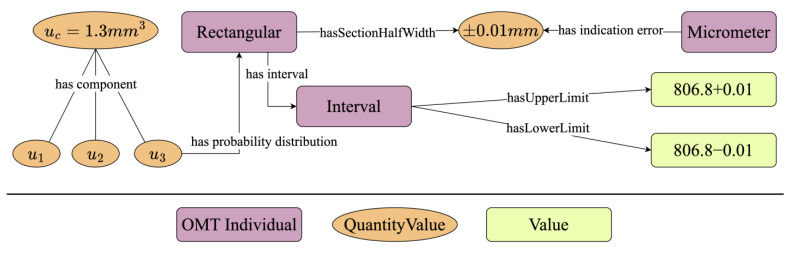
The rectangular distribution of $u_{3}$.

**Figure 6 sensors-24-03989-f006:**
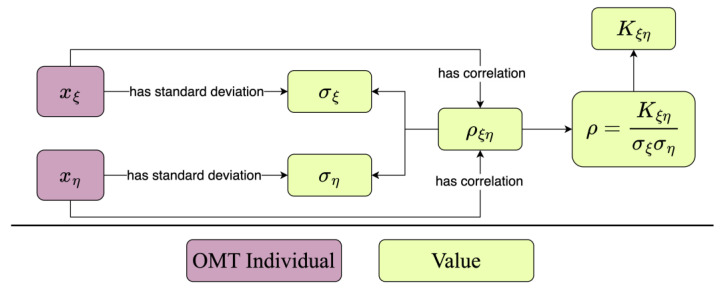
Expression of correlation coefficient.

**Figure 7 sensors-24-03989-f007:**
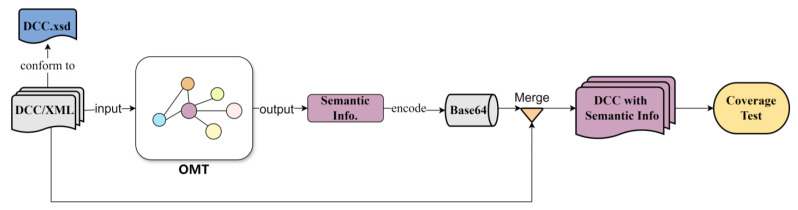
The application process of OMT on Digital Calibration Certificate (DCC).

**Table 1 sensors-24-03989-t001:** Ontologies for units and quantities.

Ontology	Scope of the Ontology
QUDT	Units of Measure, Quantity Kinds, Dimensions, and Data Types
QUDV	Quantities, Units, Dimensions, and Values
OM	Units, Quantities, Measurements, and Dimensions
vim ontology	Units, Quantity Kinds, Dimensions, and Quantity Value
SI Reference Point	Units, Prefixes, Decisions, Constants, and Quantities

**Table 2 sensors-24-03989-t002:** The inheritance hierarchy of classes.

Class	Subclass	Subsubclass
Measuring Instrument	Material Measure	-
	Indicating Measuring Instrument	-
	Displaying Measuring Instrument	-
Measurement Parameter	Measurement Uncertainty	Combined Standard Uncertainty
		Definitional Uncertainty
		Expanded Uncertainty
		Standard Uncertainty
		Relative Standard Uncertainty
	Measurement Precision	Intermediate Measurement Precision
		Reproducibility
		Measurement Repeatability
	Measurement Error	Systematic Measurement Error
		Random Measurement Error
Measured Object	-	-
Measurement Method	-	-
Operating Condition	Operating Location	-
	Temperature Condition	-
	Humidity Condition	-
Measurement Model	-	-
Measurement Procedure	-	-
Measuring System	-	-
Measurement Result	-	-

**Table 3 sensors-24-03989-t003:** Main data properties in OMT.

Data Property	Domain	Range
has value	Quantity Value	xsd:double
has coverage factor	Expanded Uncertainty	xsd:decimal
has coverage probability	Expanded Uncertainty	xsd:double
has coverage interval	Expanded Uncertainty	omt:MarkdownText
has measurement principle	Measuring System	omt:MarkdownText
has measurement function	Measurement Model	omt:LatexText
has detection limit	Measurement Result	xsd:double

**Table 4 sensors-24-03989-t004:** Key object properties in OMT.

Object Property	Domain	Range
has unit	Quantity Value	Unit
has prefix	Quantity Value	SIPrefix
has Type A component	Measurement Uncertainty	Quantity Value
has Type B component	Measurement Uncertainty	Quantity Value
has operating condition	Measuring System	Operating Condition
has deviation	Measurement Result	Quantity Value
has measurement uncertainty	Measurement Result	Measurement Uncertainty
has precision	Measurement Result	Measurement Precision
has measurement error	Measurement Result	Measurement Error
has instrumental uncertainty	Measuring Instrument	Measurement Uncertainty
has resolution	Measuring Instrument	Quantity Value
has intrinsic error	Measuring Instrument	Measurement Error
has indication	Measuring Instrument	Quantity Value
has indication error	Measuring Instrument	Measurement Error

**Table 5 sensors-24-03989-t005:** Constraints of core concepts.

Class	Constraints
QuantityValue	has value exactly 1 xsd:double
	has prefix max 1 SIPrefix
	has unit exactly 1 Unit
MeasurementResult	has measuring device min 1 MeasurementInstrument
	has measurand quantity value min 1 QuantityValue
	has measurand min 1 QuantityKind
	has model max 1 MeasurementModel
	has some measurement uncertainty some MeasurementUncertainty
MeasurementUncertainty	has uncertainty component some (TypeAUncertaintyComponent or TypeBUncertaintyComponent)
	has coverage factor max 1 xsd:double
MeasurementInstrument	has indication some QuantityValue
	has instrumental uncertainty exactly 1 MeasurementUncertainty
	cause some UncertaintyComponent
MeasurementModel	has input some QuantityKind
	has output some QuantityKind
	has measurement function min 1 LatexText
QuantityKind	has quantity value exactly 1 QuantityValue

**Table 6 sensors-24-03989-t006:** SPARQL query results for uncertainty component sources.

Component	Source	QuantityValue	Value	Prefix	Unit
UComponent_1	Height	MeasuredHeight	10.11	milli	meter
UComponent_2	Diameter	MeasuredDiameter	10.08	milli	meter

**Table 7 sensors-24-03989-t007:** Results of the coverage testing.

Domain	Concepts	Covered	Miss	Coverage
Flow Rate	341	315	26	92.38%
Mechanics	336	311	25	92.56%
Mass	435	410	25	94.25%
Power	300	274	26	91.33%
Electricity and Magnetism	412	387	25	93.93%
Length	574	549	25	95.64%
Radiation	370	345	25	93.24%
Density	265	240	25	90.57%
Time and Frequency	530	503	27	94.91%
Temperature	458	433	25	94.54%

## Data Availability

The data presented in this study are available on request from the corresponding author.

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
