# Peer review of "Design and Implementation of an Ontology for Measurement Terminology in Digital Calibration Certificates"

_sensors, 2024, doi:10.3390/s24123989_

Round 1
Reviewer 1 Report
Comments and Suggestions for Authors
The authors provide a clear description of the digitalization matter in metrology and clearly state the research gap: "the lack of a comprehensive ontology-based vocabulary that spans the broader measurement field".
To fill this gap, the authors propose the so-called Ontology for Measurement Terminology (OMT).
Overall, the manuscript is well-written; albeit, the meaning of the acronym "FAIR" ('Findable, Accessible, Interoperable, and Reusable') is missing in page 2, line 52. Please include it. The figures and tables are well-designed, thus contributing to the readability of the text.
Conceptually, the manuscript does not provide any information on the probability distribution used to obtain the standard uncertainty. The same applies to the covariances associated with input estimates (if any). Are they considered in the proposed OMT?
In fact, in the measurement process depicted in Figure 4, if the same micrometer is used to measure both the cylinder diameter and height, these estimates are correlated.
Therefore, I strongly recommend that the authors include some discussion on e.g. 'probability distribution' and 'correlation', if they are considered in the proposed OMT (and how). If not, the discussion should conveniently address this limitation.
Reviewer 2 Report
Comments and Suggestions for Authors
- the abstract is clear
- Introduction is fine
- In section II authors might improve the description of the added value of the proposed ontology and why the existing ones are not suitable.
- Section III. Please discuss the flexibility of the considered structure and its modification after its implementation.
- Fig.3 could be better organised to increase readability.
- Please describe real case applications and the practical benefits compared the current situation.
- How the approach could be further improved?
- What is the economic impact due to the implementation of the approach?
Comments on the Quality of English LanguageMinor review
Round 2
Reviewer 1 Report
Comments and Suggestions for Authors
No comments, as my recommendation is to accept the manuscript in present form.